# Factors associated with dyslipidemia among healthcare workers in a COVID-19-designated hospital in Hangzhou, Zhejiang, China: A retrospective cohort study from 2019 to 2022

Zhongbao Zuo[1‡], Lan Yu[2‡], Chunli Yang[3], Miaochan Wang[1], Jing Wu[1], Chengjiang Tao[4], Xiaofei Chen[5], Ruihua Kang[6], Shourong Liu[7], Jinsong Huang[7], Aifang Xu[1]*

1 Department of Clinical Laboratory, Hangzhou Xixi Hospital, Zhejiang, China, 2 Department of Clinical Laboratory, Community Health Service Center of Shangtang Street, Hangzhou Gongshu District, Zhejiang, China, 3 Department of Clinical Laboratory, The 903rd Hospital of PLA, Zhejiang, China, 4 Department of Obstetrics and Gynecology, Hangzhou Xixi Hospital, Zhejiang, China, 5 Human Resources Department, Hangzhou Xixi Hospital, Zhejiang, China, 6 Department of Cancer Epidemiology, The Affiliated Cancer Hospital of Zhengzhou University & Henan Cancer Hospital, Zhengzhou, China, 7 Department of Hepatology, Hangzhou Xixi Hospital, Zhejiang, China

‡ These authors share the first authorship.
* 13616500869@163.com

## Abstract

### Background

This study investigated dyslipidemia and its relative factors among Chinese healthcare workers from 2019 to 2022.

### Method

This retrospective cohort study was conducted from 2019 to 2022. The endpoints were dyslipidemia or the end of follow-up. Univariate Cox proportional hazard regression and LASSO regression models were used to select variables, and a multivariate Cox proportional hazard regression model was constructed to explore factors associated with dyslipidemia.

### Results

67 (9.2%) medical staff members were diagnosed with dyslipidemia, 106 (14.5%) resigned from the hospital, and 558 (76.3%) kept normal lipid files. Compared with healthcare workers with previous working time <10 years, the hazard ratios (HRs) of those with 10−20 years and ≥ 20 years of working experience were 0.34 (0.18–0.64) (P = 0.001) and 0.47 (0.26–0.85) (P = 0.01); compared with 0-day frontline working time, the HR of those with ≥ 30 days frontline working time was 0.38 (0.19–0.75) (P = 0.005). The HRs of TG, HDL, LDL, TBIL and HB were 3.14 (1.65–6.01) (P < 0.001), 0.20 (0.06–0.65) (P = 0.008), 2.93 (1.70–5.05) (P < 0.001), 1.06 (1.02–1.10) (P = 0.002) and 0.98 (0.97–0.99) (P = 0.04), respectively.

**Data availability statement:** All relevant data are within the manuscript and its Supporting Information files.

**Funding:** This research was funded by Hangzhou Biomedical and Health Industry Development Support Technology Special Project (2022WJC284) and Hangzhou Medical and Health Research Project (Z20230116). The funders had no role in the study design, data collection, analysis, decision to publish, or preparation of the manuscript.

**Competing interests:** The authors have declared that no competing interests exist.

## Conclusion

Healthcare workers with high frontline working time and longer previous working time were less likely to have dyslipidemia, while healthcare workers with high levels of TG, LDL, HB, TBIL, and low levels of HDL were more likely to have dyslipidemia. Supporting healthcare workers should be a priority for policymakers and hospital administrators.

## 1. Introduction

The coronavirus disease 2019 (COVID-19) has infected more than 761 million people worldwide, with total deaths exceeding 6.8 million [1]. The Chinese government has implemented a series of interventions to prevent the pandemic, including designated hospitals for COVID-19 treatment [2], 48-h or 72-h nucleic acid detection requirements [3], strict nonpharmaceutical interventions [4,5], and vaccination [6,7]. There were only 371918 confirmed COVID-19 patients in mainland China by the end of December 14, 2022 [8] (after this day, the government has taken the strategy of being "willing to check and complete the test", and many asymptomatic infected people no longer participated in nucleic acid tests), with nearly 3 years of the fight against COVID-19 for 1.4 billion people. The COVID-19 pandemic put extraordinary pressure on frontline healthcare workers both physically and psychologically, such as an unstable work environment, strict control measures, heavy workload, stay from family, insufficient personal equipment, and infection risk.

In designated hospitals, frontline healthcare workers need to isolate themselves to treat confirmed COVID-19 patients, which could influence their physical activity and eating habits [9,10]. The consumption of fruits, fish, and vegetables decreased, while salty and sugary snacks (such as candy, potato chips, desserts, nuts, biscuits, popcorn, etc.) increased [9]. Staying away from family and friends makes it difficult for them to fall asleep at night, and being confined in a limited hotel makes it difficult for them to exercise effectively [10]. The heavy workload, missing their families, and mental problems can undermine the appetite of frontline healthcare workers [9,10]. A study [11] that included 5271 Chinese frontline healthcare workers found that 26.4% had a decreased BMI, 42.0% had a stable BMI, and 31.6% had an increased BMI during the pandemic. However, the author only described changes in BMI and ignored the normal range of BMI. Even if the BMI change of frontline healthcare workers is greater or less than 0.5 (the author defined this as the moderate increase/decrease group), their BMI is still within the normal range, making it difficult to determine whether the impact of BMI changes on the body is good or bad. Therefore, it is difficult to understand how the physical condition of Chinese healthcare workers changed during the COVID-19 pandemic.

Many studies [12,13] have demonstrated that frontline healthcare workers have burnout, anxiety, depression, and other mental health problems. Meanwhile, strict

prevention strategies such as lockdown and isolation may increase emotional eating behaviors [14], which can lead to dyslipidemia for frontline healthcare workers. Researchers have found that dyslipidemia is associated with chronic inflammation [15], type 2 diabetes [15,16], hypertension [16], and cardiovascular disease [17]. Studies found that blood LDL significantly increased and HDL significantly decreased after a lockdown in 38 cases of 60–70-year-old people [18] and 6236 general workers [19]. Frontline healthcare workers face more continued pressure and infection risks than the general population, which could affect their physical status, such as dyslipidemia. Therefore, it is important to determine the risk factors for dyslipidemia among Chinese healthcare workers in the past three years of the COVID-19 pandemic.

China has implemented strict strategies for COVID-19 for nearly three years. Previous research [12,13] emphasized the mental health of medical personnel, and few researchers either focused on medical staff for a while or focused on the general population, such as older people [18], normal workers [19], and diabetes patients [20]. This study aimed to investigate dyslipidemia and its relative factors among Chinese healthcare workers from 2019 to 2022.

## 2. Method

### 2.1. The designated hospitals of Xixi hospital

Hangzhou Xixi Hospital was an officially designated hospital for COVID-19 patients from 31/01/2020 to 14/01/2023 under China's prevention and control strategies. There were two "major events" and "one continuous theme" during the three years of fighting against COVID-19. The first "event" occurred at the beginning of COVID-19, when Hangzhou Xixi Hospital closed and was locked from 23/01/2020 to 08/03/2020 to treat confirmed patients. The second lockdown "event" was from 14/01/2022 to 28/01/2022 (there was an infected nurse, and all the staff were isolated for two weeks) and 31/01/2022 to 08/05/2022 (the COVID-19 outbreak in Hangzhou and Shanghai). Xixi Hospital was completely closed during the lockdown period of the two events and could only treat COVID-19 patients. Apart from the two large events, Xixi Hospital can provide health services to other patients. There were still confirmed COVID-19 patients who needed treatment, so a building was emptied for the treatment of COVID-19 patients (this was the "one continuous theme"). Medical staff in charge of the treatment of COVID-19 patients in Hangzhou Xixi Hospital need to work for 28 days, be isolated for 14 days, and rest for 10 days. To prevent the potential spread of the virus, all medical staff should follow the "three points one line" principle: hospital-car-hotel (hospital for work, car for transportation, hotel for rest).

### 2.2. Healthcare workers in Xixi hospital

This retrospective cohort study collected healthcare workers' physical examination results from 2019 to 2022 in Hangzhou, China, and we chose 2019 as the baseline. There were three periods: 2019–2020 means the time of medical staff from the time of the 2019 physical examination to the 2020 physical examination, and so on 2020–2021, and 2021–2022. All the data were collected between 01/05/2023 and 31/07/2023 after we got the approval of Hangzhou Xixi Hospital's institutional ethics review committee. All healthcare workers should finish the baseline examinations. The inclusion criteria were (1) 18 years or older and (2) full-time doctors, nurses, or other support workers involved in hospital administration or patient care. The exclusion criteria were as follows: (1) retired medical staff between 2019 and 2022; (2) medical staff with missing information at baseline; and (3) healthcare workers with dyslipidemia at baseline. The flowchart of this study is shown in Fig 1. Hangzhou Xixi Hospital's institutional ethics review committee approved the study (2023 Science **Ethics No. 26**). Written informed consent was not required due to the retrospective nature of this study. All the data used in this study were anonymized.

### 2.3. Outcome ascertainment

The study outcome was the occurrence of dyslipidemia, and the diagnosis criteria adopted the newly published Chinese guidelines for lipid management (2023) [21]. There were four indicators, and one or more of the indicators can be used to diagnose dyslipidemia.

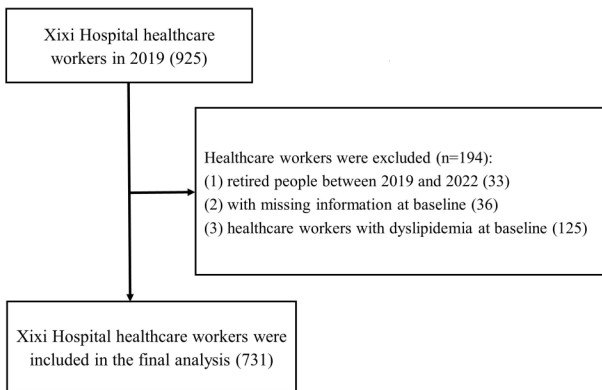

**Fig 1. Flowchart of the retrospective cohort study in Hangzhou Xixi Hospital.**

(1) Total cholesterol (TC) ≥ 6.2 mmol/L;

(2) Low-density lipoprotein cholesterol (LDL) ≥ 4.1 mmol/L;

(3) Triglyceride (TG) ≥ 2.3 mmol/L;

(4) High-density lipoprotein cholesterol (HDL) < 1.0 mmol/L.

## 2.4. Variable selection

The demographic information included gender, age, education, previous working time, department, and establishment positions (Medical staff are divided into those with establishment positions or not, and those with establishment positions tend to have high social status, decent pay, and good social welfare). Frontline working time was counted every month from Jan 2020 to Dec 2022 for all frontline healthcare workers. For medical staff diagnosed with dyslipidemia, the total frontline working time from January 2020 to the month of dyslipidemia will be recorded. For medical staff who resigned without dyslipidemia, the calculation of frontline working days started from January 2020 to the month of resignation. For medical staff who maintained normal lipid files from 2020 to 2022, the calculation of frontline working days started from January 2020 to the month of examination in 2022. For example, a female healthcare worker had normal blood lipids from 2020 to 2022, and her physical examination in 2022 was done in August, so her frontline work time was accumulated from January 2020 to August 2022.

All blood tests were performed at Hangzhou Xixi Hospital. Biochemical parameters, including triglyceride (TG), cholesterol (CHOL), high-density lipoprotein cholesterol (HDL), low-density lipoprotein cholesterol (LDL), glucose (GLU), creatinine (CREA), blood urea nitrogen (BUN), uric acid (URIC), total bilirubin (TBIL), alanine aminotransferase (ALT), aspartate aminotransferase (AST), gamma-glutamyltransferase (GGT), alkaline phosphatase (ALP), total protein (TP), and lactate dehydrogenase (LDH), were detected with a Beckman Coulter au5831 automatic biochemical analyzer. White blood cell (WBC), red blood cell (RBC), hemoglobin (HB), and platelet (PLT) were tested by an automatic blood cell analyzer.

## 2.5. Statistical analysis

The frontline working time was grouped as none (0 days), moderate (0–30 days), and high (≥ 30 days). Medical staff in charge of the treatment of COVID-19 patients in Hangzhou Xixi Hospital need to work for 28 days, be isolated for 14 days,

and rest for 10 days, and the frontline working time was recognized as 28 days. So we chose 30 days as the cutoff, which means medical staff have participated in frontline medical work one time in the moderate time group, and at least two times in the high group.

Categorical variables are shown as numbers and percentages. Skewed data are presented as medians and interquartile ranges (IQRs). Chi-squared tests for categorical variables and Kruskal–Wallis tests for skewed continuous variables were used to check the group differences.

Person weeks of follow-up were calculated from baseline (2019) to the first endpoint. The endpoint was defined as follows: dyslipidemia or the end of follow-up, whichever came first.

We checked the proportional hazard assumption and found that it did not violate this study. Univariate Cox proportional hazard regression models were constructed to explore factors associated with dyslipidemia. Then, we used the Least Absolute Shrinkage and Selection Operator (LASSO) to select variables and then built a multivariate Cox proportional hazard regression model.

Given the "three points one line" principle of the medical staff in the fight against COVID-19, we extracted the frontline working time and used the models to study the impact of frontline working time on dyslipidemia among different subgroups. In each subgroup analysis, we adjusted for all remaining variables to account for the impact of frontline working time on the occurrence of dyslipidemia.

To enhance the robustness of the model assessing the impact of frontline working time on the occurrence of dyslipidemia, three models were built. Model 1 Adjusted for age, gender, and education. Model 2 Adjusted for variables included in model 1 + previous working time, department, and establishment positions. Model 3 Adjusted for variables included in model 2 + TG, CHOL, HDL, LDL, GLU, CREA, BUN, URIC, TBIL, ALT, AST, GGT, ALP, TP, ALB, LDH, WBC, RBC, HB, and PLT.

Two-sided p values <0.05 were considered statistically significant in our study. All statistical analyses were conducted with R software (version 4.0.2, R Development Core Team 2020).

## 3. Results

### 3.1. Baseline characteristics and endpoints of the healthcare workers

The baseline characteristics of healthcare workers in total and stratified by frontline working time are presented in Table 1. The median age of the medical staff was 34 (30–41) years, 168 (23.0%) of them were male, 611 (83.6%) of them were frontline workers, 586 (80.2%) of them had establishment positions, and 421 (57.6%) of them had more than ten years of working time. Except for HDL, TBIL, CK, LDH, TP, WBC, and PLT, the remaining variables were significantly different among the three frontline working time groups (Table 1). The overall Kaplan–Meier curve stratified by demographic variables is shown in Fig 2.

There were 731 healthcare workers included in our analysis. 67 (9.2%) medical staff members were diagnosed with dyslipidemia, 106 (14.5%) medical staff members resigned from the hospital, and 558 (76.3%) medical staff members maintained a normal lipid file until 2022. Of the 67 dyslipidemia medical workers, 28 (3.8%) frontline medical personnel occurred in 2019–2020, 23 (3.2%) occurred in 2020–2021, and 16 (2.2%) occurred in 2021–2022. 65 (8.9%), 11 (1.5%), and 30 (4.1%) medical staff resigned from the hospital in 2019–2020, 2020–2021, and 2021–2022, respectively. Detailed information about the endpoint information is shown in Table 2.

### 3.2. Factors associated with dyslipidemia among healthcare workers

As shown in Table 3, gender, previous working time, frontline working time, TG, CHOL, HDL, LDL, CREA, URIC, TBIL, WBC, RBC, and HB were significantly associated with dyslipidemia in the univariate analysis. The thirteen variables in the univariate were then selected by LASSO, and 8 variables (S1 Fig) were finally selected. However, the variable

**Table 1. Baseline characteristics of healthcare workers by frontline working time groups.**

| Variables | Total (n = 731) | Frontline working time (days) | | | P value |
|---|---|---|---|---|---|
| | | None: 0 (n = 295) | Moderate: 0–30 (n = 127) | High: ≥30 (n = 309) | |
| Age, Median (IQR) | 34 (30, 41) | 34 (30, 46) | 37 (33, 43) | 32 (29, 37) | < 0.001 |
| Gender, n (%) | | | | | < 0.001 |
| Male | 168 (23.0) | 80 (27.1) | 43 (33.9) | 45 (14.6) | |
| Female | 563 (77.0) | 215 (72.9) | 84 (66.1) | 264 (85.4) | |
| Department, n (%) | | | | | < 0.001 |
| Frontline workers | 611 (83.6) | 189 (64.1) | 116 (91.3) | 306 (99) | |
| Support workers | 120 (16.4) | 106 (35.9) | 11 (8.7) | 3 (1) | |
| Previous working time, n (%) | | | | | < 0.001 |
| < 10 years | 310 (42.4) | 121 (41) | 35 (27.6) | 154 (49.8) | |
| 10–20 years | 246 (33.7) | 75 (25.4) | 55 (43.3) | 116 (37.5) | |
| ≥ 20 years | 175 (23.9) | 99 (33.6) | 37 (29.1) | 39 (12.6) | |
| Education, n (%) | | | | | < 0.001 |
| Junior college | 64 (8.8) | 43 (14.6) | 5 (3.9) | 16 (5.2) | |
| Undergraduate | 572 (78.2) | 218 (73.9) | 107 (84.3) | 247 (79.9) | |
| Postgraduate and above | 95 (13) | 34 (11.5) | 15 (11.8) | 46 (14.9) | |
| Establishment positions, n (%) | | | | | < 0.001 |
| No | 145 (19.8) | 94 (31.9) | 11 (8.7) | 40 (12.9) | |
| Yes | 586 (80.2) | 201 (68.1) | 116 (91.3) | 269 (87.1) | |
| TG, Median (IQR) | 0.94 (0.74, 1.29) | 1.01 (0.76, 1.32) | 1.02 (0.78, 1.4) | 0.87 (0.71, 1.19) | < 0.001 |
| CHOL, Median (IQR) | 4.72 (4.17, 5.21) | 4.77 (4.23, 5.3) | 4.83 (4.36, 5.38) | 4.59 (4.01, 5.03) | < 0.001 |
| HDL, Median (IQR) | 1.41 (1.24, 1.62) | 1.43 (1.23, 1.62) | 1.39 (1.22, 1.6) | 1.42 (1.25, 1.62) | 0.32 |
| LDL, Median (IQR) | 2.4 (2.03, 2.76) | 2.47 (2.12, 2.82) | 2.51 (2.17, 2.8) | 2.27 (1.94, 2.67) | < 0.001 |
| TBIL, Median (IQR) | 12.2 (9.6, 16.0) | 12.1 (9.7, 15.9) | 13.2 (9.7, 16.2) | 11.9 (9.3, 15.6) | 0.24 |
| ALT, Median (IQR) | 14 (11, 21) | 14 (11, 21) | 15 (11, 25) | 13 (10, 18) | 0.003 |
| ALP, Median (IQR) | 67 (55, 82) | 71 (58, 87) | 66 (55, 81) | 64 (53, 79) | < 0.001 |
| GGT, Median (IQR) | 17 (13, 22) | 18 (14, 23) | 18 (13, 24) | 16 (13, 20) | 0.04 |
| CK, Median (IQR) | 72 (53, 99) | 72 (55, 101) | 75 (54, 103) | 70 (52, 93) | 0.10 |
| AST, Median (IQR) | 20 (17, 23) | 20 (17, 24) | 21 (18, 24) | 19 (17, 23) | 0.004 |
| LDH, Median (IQR) | 159 (143, 177) | 158 (140, 179) | 160 (143.5, 174) | 159 (144, 177) | 0.62 |
| CREA, Median (IQR) | 61 (55, 71) | 63 (56, 73) | 63 (56, 76) | 60 (55, 67) | < 0.001 |
| BUN, Median (IQR) | 4.6 (3.8, 5.3) | 4.8 (3.95, 5.45) | 4.6 (3.85, 5.5) | 4.5 (3.7, 5.1) | 0.02 |
| URIC, Median (IQR) | 282 (243, 334) | 286 (246, 338) | 293 (247, 346) | 274.1 (235, 319) | 0.005 |
| TP, Median (IQR) | 75.4 (72.6, 78.3) | 75.2 (72.7, 78.1) | 75.3 (72.4, 77.7) | 75.9 (72.5, 79.1) | 0.309 |
| GLU, Median (IQR) | 5.01 (4.78, 5.27) | 5.05 (4.83, 5.36) | 5 (4.76, 5.34) | 4.98 (4.75, 5.21) | 0.02 |
| WBC, Median (IQR) | 5.68 (4.89, 6.66) | 5.67 (4.74, 6.62) | 5.62 (4.97, 6.86) | 5.7 (4.95, 6.57) | 0.77 |
| RBC, Median (IQR) | 4.59 (4.35, 4.95) | 4.64 (4.37, 5.03) | 4.68 (4.4, 5.12) | 4.54 (4.34, 4.81) | 0.001 |
| HB, Median (IQR) | 135 (127.5, 145) | 136 (129, 146) | 138 (127, 151.5) | 133 (126, 141) | < 0.001 |
| PLT, Median (IQR) | 241 (207, 280) | 246 (212.5, 281) | 232 (201.5, 267) | 241 (206, 280) | 0.32 |

Abbreviations: TG, triglyceride; CHOL, cholesterol; HDL, high-density lipoprotein; LDL, Low-Density Lipoprotein; GLU, glucose; CREA, creatinine; BUN, blood urea nitrogen; URIC, uric acid; TBIL, total bilirubin; ALT, alanine aminotransferase; AST, aspartate aminotransferase; GGT, gamma-glutamyltransferase; ALP, alkaline phosphatase; TP, total protein; ALB, albumin; LDH, lactate dehydrogenase; WBC, white blood cell; RBC, red blood cell; HB, hemoglobin; PLT, platelet.

 

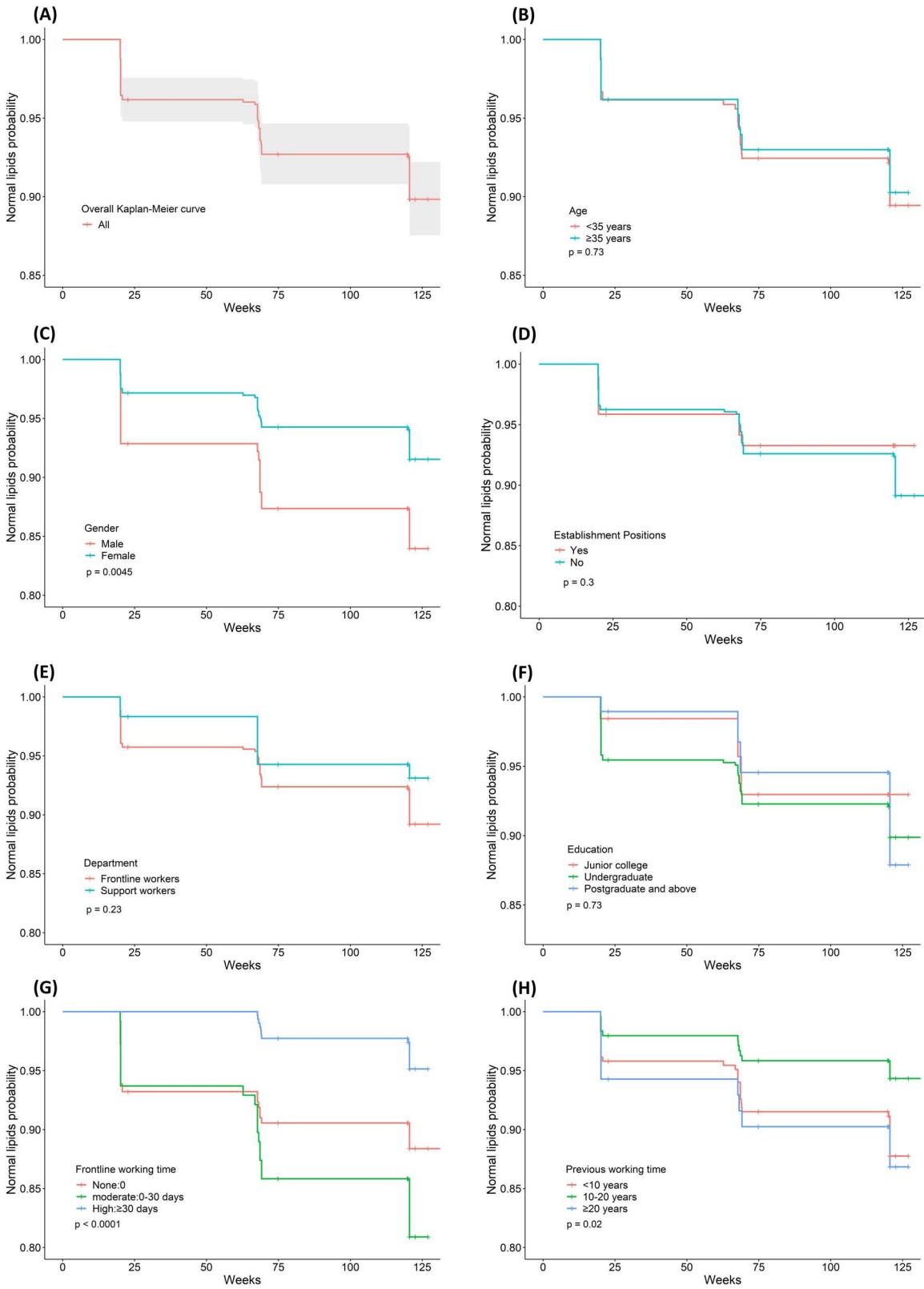

**Fig 2. Kaplan–Meier curves for normal lipid probability among the demographic variables. (A)** Overall Kaplan–Meier curves; **(B)** Kaplan–Meier curves with age; **(C)** Kaplan–Meier curves with gender; **(D)** Kaplan–Meier curves with establishment positions; **(E)** Kaplan–Meier curves with department; **(F)** Kaplan–Meier curves with education; **(G)** Kaplan–Meier curves with frontline working time; **(H)** Kaplan–Meier curves with previous working time.

**Table 2. Three outcomes of the 731 healthcare workers from 2019-2022.**

| Characteristic | N (%) |
|---|---|
| Total | 731 (100.0%) |
| Dyslipidemia | 67 (9.2%) |
| 2019-2020 | 28 (3.8%) |
| 2020-2021 | 23 (3.2%) |
| 2021-2022 | 16 (2.2%) |
| Resignation | 106 (14.5%) |
| 2019-2020 | 65 (8.9%) |
| 2020-2021 | 11 (1.5%) |
| 2021-2022 | 30 (4.1%) |
| End of the follow-up | 558 (76.3%) |

of WBC was deleted because of no significant change of the Akaike information criterion (AIC). In the multivariate Cox proportional hazard regression model, previous working time, frontline working time, TG, HDL, LDL, HB, and TBIL were independently associated with dyslipidemia. Compared with healthcare workers with previous working time <10 years, the hazard ratios (HRs) of those with 10−20 years and ≥ 20 years of working experience were 0.34 (0.18–0.64) (P = 0.001) and 0.47 (0.26–0.85) (P = 0.01); compared with 0-day frontline working time, the HR of those with ≥ 30 days frontline working time was 0.38 (0.19–0.75) (P = 0.005). The HRs of TG, HDL, LDL, TBIL and HB were 3.14 (1.65–6.01) (P < 0.001), 0.20 (0.06–0.65) (P = 0.008), 2.93 (1.70–5.05) (P < 0.001), 1.06 (1.02–1.10) (P = 0.002) and 0.98 (0.97–0.99) (P = 0.04), respectively.

### 3.3. Associations of frontline working time with dyslipidemia among different subgroups

Table 4 shows the associations of frontline working time with dyslipidemia. Compared with 0-day frontline working time, the HRs of the three models for medical staff with frontline working time ≥ 30 days were significantly below 1. However, the HRs of 0–30 days frontline working time for the three models were not statistically significant.

Fig 3 shows the associations between frontline working time and dyslipidemia stratified by gender, age, establishment positions, department, and previous working time. Compared with the 0-day frontline working time, ≥ 30 days frontline working time had negative effects on dyslipidemia in the subgroups of female, < 35 years, ≥ 35 years, establishment positions (Yes), frontline workers, previous working time <10 years, and previous working time ≥ 10 years.

### Discussion

This retrospective study aimed to investigate dyslipidemia and its relative factors among Chinese healthcare workers from 2019 to 2022. We included 731 employees from Hangzhou Xixi Hospital, which consisted of frontline medical personnel (doctors and nurses) and hospital support personnel (administrative staff and other support workers to maintain normal service). The main results showed that previous working time, frontline working time, TG, HDL, LDL, HB, and TBIL were independently associated with dyslipidemia. Compared with 0-day frontline working time, the HRs for medical staff with frontline working time of ≥ 30 days were significantly below 1 after adjusting for other covariates. However, we found that

**Table 3. Factors associated with Dyslipidemia among healthcare workers in Hangzhou, Zhejiang, China.**

| Variables | Crude HR (95%CI) | P-value | Adjusted HR (95%CI) | P-value |
|---|---|---|---|---|
| Age | 1.01 (0.98-1.04) | 0.58 | | |
| Gender, n (%) | | | | |
| Male | 1 [Reference] | | | |
| Female | 0.49 (0.29-0.81) | 0.005 | | |
| Education, n (%) | | | | |
| Junior college | 1 [Reference] | | | |
| Undergraduate | 1.54 (0.48-4.94) | 0.47 | | |
| Postgraduate and above | 1.68 (0.46-6.09) | 0.43 | | |
| Department, n (%) | | | | |
| Frontline workers | 1 [Reference] | | | |
| Support workers | 0.62 (0.28-1.36) | 0.23 | | |
| Establishment positions, n (%) | | | | |
| No | 1 [Reference] | | | |
| Yes | 1.45 (0.72-2.93) | 0.29 | | |
| Previous Working time, n (%) | | | | |
| <10 years | 1 [Reference] | | 1 [Reference] | |
| 10–20 years | 0.45 (0.24-0.85) | 0.01 | 0.34 (0.18-0.64) | <0.001 |
| ≥20 years | 0.54 (0.17-1.77) | 0.31 | 0.47 (0.26-0.85) | 0.01 |
| Frontline working time (days) | | | | |
| None: 0 | 1 [Reference] | | 1 [Reference] | |
| Moderate: 0–30 | 1.57 (0.91-2.70) | 0.10 | 1.51 (0.87-2.63) | 0.15 |
| High: ≥30 | 0.35 (0.19-0.67) | 0.001 | 0.38 (0.19-0.75) | 0.005 |
| TG | 5.70 (3.41-9.52) | <0.001 | 3.14 (1.65-6.01) | <0.001 |
| CHOL | 2.25 (1.57-3.23) | <0.001 | | |
| HDL | 0.08 (0.02-0.24) | <0.001 | 0.20 (0.06-0.65) | 0.008 |
| LDL | 4.67 (2.82-7.76) | <0.001 | 2.93 (1.70-5.05) | <0.001 |
| GLU | 1.37 (0.88-2.16) | 0.17 | | |
| CREA | 1.02 (1.01-1.03) | 0.03 | | |
| BUN | 1.21 (0.99-1.48) | 0.06 | | |
| URIC | 1.006 (1.003-1.009) | <0.001 | | |
| TBIL | 1.05 (1.01-1.09) | 0.01 | 1.06 (1.02-1.10) | 0.002 |
| ALT | 1.01 (0.99-1.02) | 0.39 | | |
| AST | 1.01 (0.98-1.03) | 0.53 | | |
| GGT | 1.01 (0.99-1.02) | 0.07 | | |
| ALP | 1.00 (0.99-1.01) | 0.32 | | |
| TP | 1.01 (0.95-1.06) | 0.97 | | |
| LDH | 1.01 (0.99-1.02) | 0.19 | | |
| WBC | 1.16 (1.02-1.31) | 0.02 | | |
| RBC | 1.86 (1.12-3.08) | 0.02 | | |
| HB | 1.02 (1.00-1.03) | 0.04 | 0.98 (0.97-0.99) | 0.04 |
| PLT | 1.00 (0.99-1.01) | 0.23 | | |

Abbreviations: TG, triglyceride; CHOL, cholesterol; HDL, high-density lipoprotein; LDL, Low-Density Lipoprotein; GLU, glucose; CREA, creatinine; BUN, blood urea nitrogen; URIC, uric acid; TBIL, total bilirubin; ALT, alanine aminotransferase; AST, aspartate aminotransferase; GGT, gamma-glutamyltransferase; ALP, alkaline phosphatase; TP, total protein; LDH, lactate dehydrogenase; WBC, white blood cell; RBC, red blood cell; HB, hemoglobin; PLT, platelet.

**Table 4. Associations of frontline working time with Dyslipidemia.**

| Frontline working time | Number | Cases/person-weeks | Hazard ratio (95%CI) | | |
|---|---|---|---|---|---|
| | | | Model 1[a] | Model 2[b] | Model 3[c] |
| 0 | 295 | 30/26729 | 1 [Reference] | 1 [Reference] | 1 [Reference] |
| 0-30 days | 127 | 23/13657 | 1.50 (0.87-2.59) | 1.09 (0.61-1.95) | 1.03 (0.55-1.94) |
| ≥ 30 days | 309 | 14/36867 | 0.36 (0.19-0.68)* | 0.26 (0.13-0.51)* | 0.17 (0.08-0.36)* |

*P<0.001.

[a]Adjusted for age, gender, and education.

[b]Adjusted for variables included in model 1+previous working time, department, and establishment positions.

[c]Adjusted for variables included in model 2+TG, CHOL, HDL, LDL, GLU, CREA, BUN, URIC, TBIL, ALT, AST, GGT, ALP, TP, LDH, WBC, RBC, HB, and PLT.

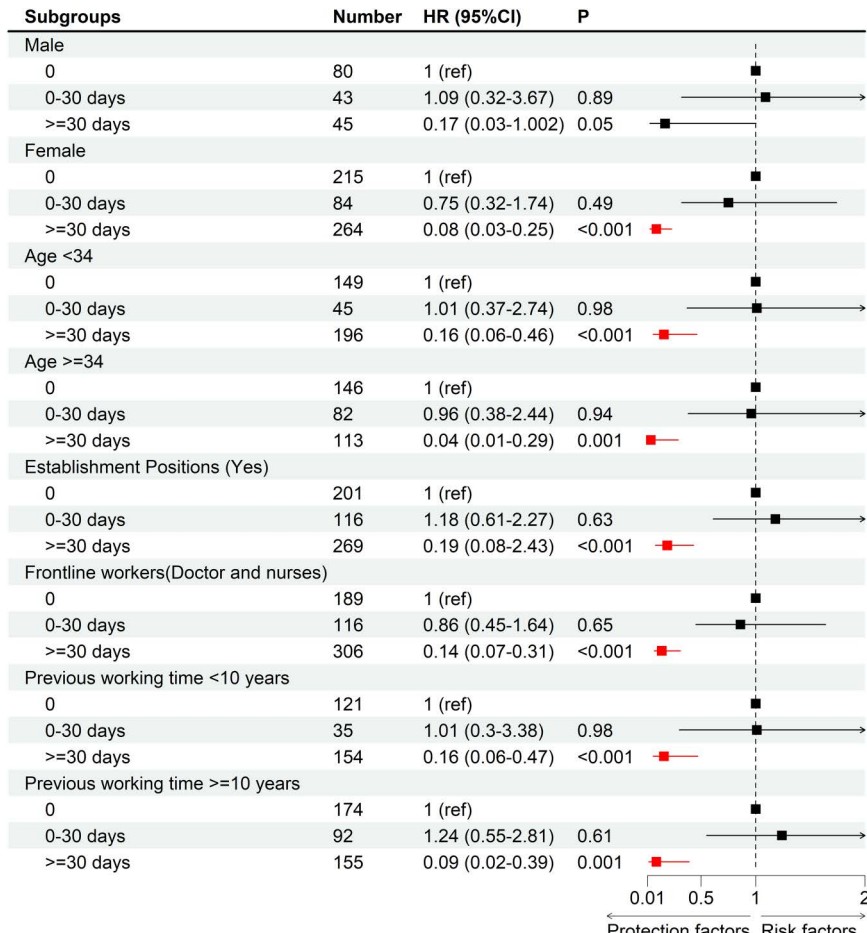

**Fig 3. Associations between dyslipidemia and frontline working time were stratified by gender, age, establishment positions, department, and previous working time.** HRs were adjusted for age (not adjusted in subgroup analysis by age), gender (not adjusted in subgroup analysis by gender), education, previous working time (not adjusted in subgroup analysis by previous working time), department (not adjusted in subgroup analysis by department), establishment positions (not adjusted in subgroup analysis by establishment positions), TG, CHOL, HDL, LDL, GLU, CREA, BUN, URIC, TBIL, ALT, AST, GGT, ALP, TP, LDH, WBC, RBC, HB, and PLT.

106 medical personnel resigned from the hospital during the follow-up, which should be paid more attention to by the government's relevant departments.

To our knowledge, this is the first study to investigate dyslipidemia and its relative factors in Chinese healthcare workers. Interestingly, compared with medical staff with a 0-day frontline working time, healthcare workers with a high (≥ 30 days) frontline working time were less likely to experience dyslipidemia, while healthcare workers with a moderate (0–30 days) frontline working time were not significantly different. There were three possible explanations. Firstly, the COVID-19 pandemic put extraordinary pressure on healthcare workers both physically and psychologically [22,23]. The heavy workload will keep healthcare workers physically active, thus leading to an increase in energy consumption. However, missing families, anxiety, depression, and other mental problems can undermine the appetite of healthcare workers and thus lead to a decrease in energy intake [9,10]. Our results suggest that the outcome of the paradox was that mental health may have a more significant impact on energy intake. A study [24] from Hubei Province conducted from March 19 to April 1, 2020, found that 22.9% of healthcare workers reported weight loss, and more than 50% of healthcare workers asked for a balanced diet, including more coarse grains, vegetables, nuts, fruits, and soybeans. Secondly, most healthcare workers who entered the frontline were more physically and psychologically healthy, so there may be selection bias. Research [25] also found that medical staff who maintain their self-regulatory eating behavior are more likely to be stress-free. Thirdly, those medical personnel who have not entered the front line have more time with families and a more relaxed environment to consume delicious food, which to some extent can also lead to a higher lipid profile. Therefore, this study found that medical staff with a high (≥ 30 days) frontline working time have a lower risk of developing dyslipidemia.

We also found that previous working time had negative effects on dyslipidemia. Compared with healthcare workers with <10 years of working time, the HRs of those with 10−20 years and ≥ 20 years of working time were 0.37 (0.19–0.71) (P = 0.003) and 0.49 (0.28–0.89) (P = 0.02), respectively. As discussed above, healthcare workers need to isolate themselves to treat confirmed COVID-19 patients, putting them at greater risk for physical health and worsening mental health. COVID-19 prevention and control may lead to more conflicts between healthcare workers and their families. Research [26] found that healthcare workers with 1−5, 6−10, and ≥11 years of service had 1.78 (1.61–1.96), 2.31 (2.10–2.53), and 2.43 (2.18–2.69) folds to experience work-family conflict compared with <1 year of service, which in turn made them better at handling these conflicts under China's prevention and control strategy. Healthcare workers with longer working service times were able to maintain emotional stability and adjust their state through yoga [27] and meditation [28]. Healthcare workers with short service years face more challenges and pressure in clinical work, which leads to emotional eating [25,29] and thus affects their blood lipids.

We found that the baseline levels of TG, HDL, and LDL were independently associated with dyslipidemia in our study. Previous research found that lipid files such as TG, LDL, and CHOL significantly increased from pre-lockdown to post-lockdown among healthcare workers [30,31], diabetes patients [20,32], and general populations [18,19]. Medical staff with high TG, LDL, and low HDL at baseline were more likely to have dyslipidemia during the pandemic. The diagnostic criteria for dyslipidemia followed the newly published Chinese guidelines for lipid management (2023) [21], but the guidelines also proposed the concept of "slightly elevated", which means patients with CHOL ≥ 5.2 and < 6.2 mmol/L, LDL ≥ 3.4 and <4.1 mmol/L or TG ≥ 1.7 and <2.3 mmol/L. Here, in our study, we did find that a higher normal lipid file at baseline was more likely to result in dyslipidemia in future follow-ups. Among the 67 healthcare workers with dyslipidemia, 35 (52.2%) had "slightly elevated" at baseline. Therefore, hospital administrators and policymakers should pay more attention to healthcare workers with "slightly elevated" lipids at baseline to prevent dyslipidemia. "Slightly elevated" of blood lipids but still within the normal range, so there is no need to take lipid-lowering drugs temporarily. However, elevated levels of LDL are an independent risk factor for atherosclerotic cardiovascular disease (ASCVD). The higher the LDL levels, the greater the risk of cardiovascular events [33], such as coronary heart disease, myocardial infarction, and stroke. While TG is not an independent risk factor for coronary heart disease, the presence of elevated TG levels in combination with other abnormalities, such as high LDL and low HDL, significantly increases the risk of cardiovascular disease. Healthcare workers

can lower blood lipids by changing their lifestyle habits, such as paying attention to a low-salt and low-fat diet, engaging in regular exercises such as running and skipping rope, and ensuring sufficient sleep time and good sleep quality. These measures can help healthcare workers control their blood lipid levels. If not taken seriously, blood lipids can escalate from "slightly elevated" to dyslipidemia and even hyperlipidemia, which significantly increases the risk of cardiovascular and cerebrovascular diseases.

China has achieved significant accomplishments in COVID-19 prevention and control, and Xixi Hospital has also done a great job in the past three years, with only one nurse infected with COVID-19. However, great achievements often come with great sacrifices. Among 287 Chinese nurses from 11 COVID-19-designated hospitals, researchers [34] found that 60.54% of the relationship between job withdrawal and risk perception of COVID-19 was mediated by work–family conflict. Frontline healthcare workers in the designated hospitals would confront more work-family conflicts than in the non-COVID-19 era, and Chinese people have a strong sense of family rather than work. A cross-sectional study [35] including 2014 frontline nurses from 13 February to 24 February 2020 in Wuhan, China, investigated their willingness to work, and the results showed that only 64 (3.2%) cases expressed their unwillingness to work. Recently, a cross-sectional study [36] from May 1, 2022, to June 1, 2022, in Jiangsu, China, found that 151 (55.93%) doctors and nurses expressed their resignation ideas in the questionnaire. However, our study found that 106 (14.5%) medical staff resigned from Hangzhou Xixi Hospital over the past three years. Among them, 65 and 30 medical personnel resigned from the hospital in 2019–2020 and 2021–2022, respectively, which coincidentally corresponded to two major quarantines in Hangzhou Xixi Hospital, and the quarantine measures indeed intensified the resignation of the medical staff. Therefore, supporting healthcare workers should be a priority for policymakers and hospital administrators. According to our study, we have three suggestions for policymakers and hospital administrators in the future. First, strengthen the education on mental health for medical personnel. Frontline medical workers may face many pressures that often lead to anxiety, depression, and other emotions. Therefore, it is necessary to strengthen their mental health and guide them to understand their emotions, and if necessary, professional psychological counseling services can be provided to medical personnel to promptly solve psychological problems and avoid worsening. We can use online mental health education and online psychological counseling services through communication programs, such as WeChat, Weibo, and TikTok, which have been widely used during the outbreak for medical staff and have shown a good reaction [37]. Second, provide stable compensation and establish good benefits. It is necessary to provide them with stable salaries and good welfare benefits so that they have the courage and motivation to do their work. Third, reduce the working hours of medical staff, give them more rest, and give them time to spend with families and friends.

This study has several limitations. Firstly, Hangzhou Xixi Hospital is a designated hospital in Zhejiang, China, and this study cannot represent the national situation. Future studies should include more designated hospitals and demonstrate the national level of physical changes among frontline medical workers during the past three years of fighting against COVID-19. Secondly, our study omitted several critical assessments for healthcare workers, including blood pressure, body mass index (BMI), and waist circumference, due to the substantial number of missing values associated with these measurements. Medical staff have a better understanding of their health status and thus did not undergo such tests during the normal yearly physical examination. Thirdly, some other important variables, such as the psychological status, lifestyle factors, and medication used by the medical staff, should be included in future research.

In conclusion, healthcare workers with high (≥ 30 days) frontline working time and longer previous working time were less likely to have dyslipidemia, while healthcare workers with high levels of TG, LDL, HB TBIL and low levels of HDL were more likely to have dyslipidemia. Although China has achieved significant accomplishments in COVID-19 prevention and control, supporting healthcare workers should be a priority for policymakers and hospital administrators.

## Supporting information

**S1 Fig. LASSO Cox regression plot.** (A) Plot of partial likelihood deviance (Each color curve represents the LASSO coefficient profile of a variable against the Log (λ) sequence.); (B) plot of LASSO coefficient profiles (The values above the figure represent the numbers of variables included in the model, with the corresponding λ shown on the x-axis; λ: lambda). (JPG)

**S1 File. Part of the original data and its analysis code.**
(ZIP)

## Acknowledgments

We would like to thank all the healthcare workers responsible for COVID-19 relative works from Hangzhou Xixi Hospital.

## Author contributions

**Conceptualization:** Shourong Liu, Jinsong Huang, Aifang Xu.

**Data curation:** Zhongbao Zuo, Lan Yu, Chunli Yang, Chengjiang Tao, Xiaofei Chen.

**Formal analysis:** Miaochan Wang, Ruihua Kang.

**Investigation:** Chunli Yang.

**Methodology:** Shourong Liu.

**Software:** Zhongbao Zuo.

**Supervision:** Jing Wu, Jinsong Huang, Aifang Xu.

**Validation:** Miaochan Wang, Jing Wu.

**Writing – original draft:** Zhongbao Zuo, Lan Yu.

**Writing – review & editing:** Zhongbao Zuo, Chunli Yang, Aifang Xu.

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
