## [Decision Letter · Decision Letter 0]

PONE-D-24-13033Factors associated with dyslipidemia among healthcare workers in a COVID-19-designated hospital in Hangzhou, Zhejiang, China: a retrospective cohort study from 2019 to 2022PLOS ONE

Dear Dr. Xu,

Thank you for submitting your manuscript to PLOS ONE. After careful consideration, we feel that it has merit but does not fully meet PLOS ONE’s publication criteria as it currently stands. Therefore, we invite you to submit a revised version of the manuscript that addresses the points raised during the review process.

We look forward to receiving your revised manuscript.

Kind regards,

Tariq Jamal Siddiqi

Academic Editor

PLOS ONE

Journal Requirements:

3. In the online submission form, you indicated that Data cannot be shared publicly because of some limitations to the public, and proper request for the data can contact the corresponding author

Reviewers' comments:

Reviewer's Responses to Questions

**Comments to the Author**

1. Is the manuscript technically sound, and do the data support the conclusions?

Reviewer #1: Yes

Reviewer #2: Yes

2. Has the statistical analysis been performed appropriately and rigorously? 

Reviewer #1: Yes

Reviewer #2: Yes

3. Have the authors made all data underlying the findings in their manuscript fully available?

Reviewer #1: Yes

Reviewer #2: Yes

4. Is the manuscript presented in an intelligible fashion and written in standard English?

Reviewer #1: Yes

Reviewer #2: Yes

5. Review Comments to the Author

**Reviewer #1:**  - in the paper, authors present data on BMI fluctuations among Chinese frontline healthcare workers during the COVID-19 pandemic. However, the statement "Therefore, it is difficult to understand how the physical condition changed among Chinese healthcare workers during the COVID-19 pandemic" contradicts the provided results. The link between the data and this conclusion is unclear. Please clarify and provide more specific details to support this statement.

- connection between increased infection risk and the development of dyslipidemia is still unclear. In my opinion the authors should provide a brief explanation or establish a clearer link to help readers better understand this relationship.

- rationale behind grouping the front-line working time under none, moderate and high is not clearly explained in the text. It would be helpful if the authors could provide a reference to previous literature or give an explanation for these thresholds.

- the authors have provided recommendations for policymakers and hospital administrators, their suggestions aren’t specific. To strengthen the discussion, it would be beneficial for them to consider more specific interventions. For instance, they could explore the implementation of online mental health education programs with communication modules and online psychological self-help intervention systems (PMID: 32085841).

- please provide references to support the explanation for the results regarding front line working time and dyslipidemia

**Reviewer #2:**  Zuo et al. investigated the prevalence and determinants of dyslipidemia among healthcare workers during the COVID-19 pandemic in "Factors associated with dyslipidemia among healthcare workers in a COVID-19-designated hospital in Hangzhou, Zhejiang, China: a retrospective cohort study from 2019 to 2022". The study found that healthcare workers with significant frontline working time and longer previous working time were less likely to develop dyslipidemia. I believe the manuscript could be further improved by incorporating the following edits:

1. Page 9, Lines 62-67: Please provide a reference to support these statements.

2. Page 10, Lines 80-82: Avoid using vague and imprecise terms like “normal people”. Instead, it would be better to compare front line workers to specified reference groups such as the general population or other relevant cohort to provide a clearer context for comparison.

3. Page 13, Line 134: It is unclear what the authors mean by the term “authorized strength” in the context of the study. Are they referring to staffing levels, qualifications or something else. Please define this term when first mentioned in the text to avoid any ambiguity.

4. Page 13, Lines 137-140: The use of the phrase “month of examination of 2022’ when evaluating frontline working time is very vague. To improve transparency and avoid any confusion, the authors should clearly explain how they calculated the time period for medical staff with normal lipid levels.

5. Page 15, Line 180: Replace the period in ‘(30.41)’ with comma when reporting median value.

6. Pages 22-23, Lines 288-290: When citing the study on work-family conflict, please provide specific results, such as percentages or statistical findings, this will help the readers better understand the comparison and improve the validity of the discussion.

7. Page 23, Line 303: Although the writers have briefly mentioned the concept of "edge elevation" in the discussion, they have not provided enough contextual information on its significance. Adding more details about the concept, including its importance and relevance to understanding the results, would

improve the clarity of the discussion.

6. PLOS authors have the option to publish the peer review history of their article (what does this mean? ). If published, this will include your full peer review and any attached files.

**Do you want your identity to be public for this peer review?** For information about this choice, including consent withdrawal, please see our Privacy Policy .

Reviewer #1: No

Reviewer #2: No

---

## [Author Response · Author response to Decision Letter 1]

20 Jul 2024

Reviewer #1: - in the paper, authors present data on BMI fluctuations among Chinese frontline healthcare workers during the COVID-19 pandemic. However, the statement "Therefore, it is difficult to understand how the physical condition changed among Chinese healthcare workers during the COVID-19 pandemic" contradicts the provided results. The link between the data and this conclusion is unclear. Please clarify and provide more specific details to support this statement.

Answer: We would like to thank you for your critical and constructive comments. In the literature we cited, the author only described changes in BMI and ignored the normal range of BMI. Therefore, even if the weight change of hospital workers is greater or less than 0.5 (the author defined as a moderate increase/decrease group), their BMI is still within the normal range, making it difficult to determine whether the impact of BMI changes on the body is good or bad. That's why we say “Therefore, it is difficult to understand how the physical condition changed among Chinese healthcare workers during the COVID-19 pandemic”, We also understand that this statement does cause some confusion for readers. We rephrase this paragraph into “However, the author only described changes in BMI and ignored the normal range of BMI. Even if the BMI change of frontline healthcare workers is greater or less than 0.5 (the author defined as the moderate increase/decrease group), their BMI is still within the normal range, making it difficult to determine whether the impact of BMI changes on the body is good or bad. Therefore, it is difficult to understand how the physical condition changed among Chinese healthcare workers during the COVID-19 pandemic.”

1. connection between increased infection risk and the development of dyslipidemia is still unclear. In my opinion, the authors should provide a brief explanation or establish a clearer link to help readers better understand this relationship.

Answer: We would like to thank you for your critical and constructive comments. At the beginning of the outbreak of COVID-19, the risk of medical staff infecting COVID-19 was very high, but with the protection and understanding of COVID-19, the risk of Chinese medical staff infecting COVID-19 under strict control measures was very low. This article mainly explores the risk of dyslipidemia faced by medical staff under strict control measures in China, and the connection between increased COVID-19 infection risk and the development of dyslipidemia was hard to evaluate cause only one nurse was infected with COVID-19 during the three years of strict control measures. By the way, we aim to evaluate the dyslipidemia situation among the Chinese frontline workers and its associated factors.

2. rationale behind grouping the front-line working time under none, moderate and high is not clearly explained in the text. It would be helpful if the authors could provide a reference to previous literature or give an explanation for these thresholds.

Answer: We would like to thank you for your critical and constructive comments. As we said in the manuscript, medical staff in charge of the treatment of COVID-19 patients in Hangzhou Xixi Hospital need to work for 28 days, be isolated for 14 days, and rest for 10 days, and the frontline working time was recognized as 28 days. We chose 30 days as the cutoff which means medical staff have participated in frontline medical work one time in the moderate time group, and at least two times in the high group. And we added the explanation in the Method- Statistical analysis part.

3. the authors have provided recommendations for policymakers and hospital administrators, their suggestions aren’t specific. To strengthen the discussion, it would be beneficial for them to consider more specific interventions. For instance, they could explore the implementation of online mental health education programs with communication modules and online psychological self-help intervention systems (PMID: 32085841).

Answer: We would like to thank you for your critical and constructive comments. We added your suggestion in the discussion with "We can use online mental health education and online psychological counseling services through communication programs, such as WeChat, Weibo, and TikTok, which have been widely used during the outbreak for medical staff and have shown a good reaction”.

4. please provide references to support the explanation for the results regarding front line working time and dyslipidemia

Answer: We would like to thank you for your critical and constructive comments. We added some references in the discussion and added more explanation in the discussion with “Medical staff who diagnose and treat COVID-19 patients are all volunteers who actively sign up, and they often have higher abilities to handle stress and self-regulation. Research (25) also found that medical staff who maintain their self-regulatory eating behavior are more likely to be stress-free. Therefore, this study found that medical staff with a high (≥ 30 days) frontline working time have a lower risk of developing dyslipidemia.”

Reviewer #2: Zuo et al. investigated the prevalence and determinants of dyslipidemia among healthcare workers during the COVID-19 pandemic in "Factors associated with dyslipidemia among healthcare workers in a COVID-19-designated hospital in Hangzhou, Zhejiang, China: a retrospective cohort study from 2019 to 2022". The study found that healthcare workers with significant frontline working time and longer previous working time were less likely to develop dyslipidemia. I believe the manuscript could be further improved by incorporating the following edits:

1. Page 9, Lines 62-67: Please provide a reference to support these statements.

Answer: We would like to thank you for your critical and constructive comments. The statements were from Ref 9-10, and we added them in the manuscript.

2. Page 10, Lines 80-82: Avoid using vague and imprecise terms like “normal people”. Instead, it would be better to compare front line workers to specified reference groups such as the general population or other relevant cohort to provide a clearer context for comparison.

Answer: We would like to thank you for your critical and constructive comments. We changed the “normal people” into “general people” and checked the manuscript to avoid these vague and imprecise terms.

3. Page 13, Line 134: It is unclear what the authors mean by the term “authorized strength” in the context of the study. Are they referring to staffing levels, qualifications or something else. Please define this term when first mentioned in the text to avoid any ambiguity.

Answer: We would like to thank you for your critical and constructive comments. And we explain the authorized strength in the manuscript to help readers better understand the context of medical staff. Here is the explanation: “Medical staff are divided into those with authorized strength or not, and those with authorized strength tend to have high social status, decent pay, and good social welfare.”

4. Page 13, Lines 137-140: The use of the phrase “month of examination of 2022’ when evaluating frontline working time is very vague. To improve transparency and avoid any confusion, the authors should clearly explain how they calculated the time period for medical staff with normal lipid levels.

Answer: We would like to thank you for your critical and constructive comments. We explain the definition with an example to help readers better understand. Here is the detail, “For example, a female healthcare worker had normal blood lipids from 2020 to 2022, and her physical examination in 2022 was done in August, so her frontline work time was accumulated from January 2020 to August 2022.”

5. Page 15, Line 180: Replace the period in ‘(30.41)’ with comma when reporting median value.

Answer: We would like to thank you for your attentiveness and seriousness. This is a typing error, and we have corrected it in the manuscript.

6. Pages 22-23, Lines 288-290: When citing the study on work-family conflict, please provide specific results, such as percentages or statistical findings, this will help the readers better understand the comparison and improve the validity of the discussion.

Answer: We would like to thank you for your critical and constructive comments. We rewrite this paragraph with “Research (23) found that healthcare workers with 1-5, 6-10, and ≥11 years of service had 1.78 (1.61-1.96), 2.31 (2.10-2.53), and 2.43 (2.18-2.69) folds to experience work-family conflict compared with <1 year of service, which in turn made them better at handling these conflicts under China’s prevention and control strategy.

7. Page 23, Line 303: Although the writers have briefly mentioned the concept of "edge elevation" in the discussion, they have not provided enough contextual information on its significance. Adding more details about the concept, including its importance and relevance to understanding the results, would improve the clarity of the discussion.

Answer: We would like to thank you for your critical and constructive comments. We add more details about the “edge elevation” in the following paragraph. “Edge elevation of blood lipids but still within the normal range, so there is no need to take lipid-lowering drugs temporarily. Healthcare workers can lower blood lipids by changing their lifestyle habits, such as paying attention to a low-salt and low-fat diet, engaging in regular exercises such as running and skipping rope, and ensuring sufficient sleep time and good sleep quality. These measures can help healthcare workers control their blood lipid levels. If not taken seriously, blood lipids can escalate from “edge elevation” to dyslipidemia, and even hyperlipidemia, which significantly increases the risk of cardiovascular and cerebrovascular diseases.”

---

## [Decision Letter · Decision Letter 1]

PONE-D-24-13033R1Factors associated with dyslipidemia among healthcare workers in a COVID-19-designated hospital in Hangzhou, Zhejiang, China: a retrospective cohort study from 2019 to 2022PLOS ONE

Dear Dr. Xu,

Thank you for submitting your manuscript to PLOS ONE. After careful consideration, we feel that it has merit but does not fully meet PLOS ONE’s publication criteria as it currently stands. Therefore, we invite you to submit a revised version of the manuscript that addresses the points raised during the review process. Specifically reviewers thought the proposal that high workload may have led to lower energy intake and better lipid profiles was speculative and required testing of other explanations. In addition the method of statistical analysis was also questioned. Please address these concerns in a revised manuscript.

We look forward to receiving your revised manuscript.

Kind regards,

Colin Johnson, Ph.D.

Academic Editor

PLOS ONE

Reviewers' comments:

Reviewer's Responses to Questions

**Comments to the Author**

1. If the authors have adequately addressed your comments raised in a previous round of review and you feel that this manuscript is now acceptable for publication, you may indicate that here to bypass the “Comments to the Author” section, enter your conflict of interest statement in the “Confidential to Editor” section, and submit your "Accept" recommendation.

Reviewer #2: All comments have been addressed

Reviewer #3: (No Response)

2. Is the manuscript technically sound, and do the data support the conclusions?

Reviewer #2: Yes

Reviewer #3: Partly

3. Has the statistical analysis been performed appropriately and rigorously? 

Reviewer #2: Yes

Reviewer #3: No

4. Have the authors made all data underlying the findings in their manuscript fully available?

Reviewer #2: Yes

Reviewer #3: (No Response)

5. Is the manuscript presented in an intelligible fashion and written in standard English?

Reviewer #2: Yes

Reviewer #3: No

6. Review Comments to the Author

Reviewer #2: (No Response)

Reviewer #3: The authors addressed an important issue regarding the factors associated with dyslipidemia in healthcare workers in a COVID-19- designated hospital in China. Though the topic is important but the study has major flaws. Some are mentioned in limitations by the authors as well.

Some of the issues that need to be addressed are

1. The study adjusts for several potential confounders in the multivariate Cox proportional hazard regression model (e.g., age, gender, frontline working time, biochemical parameters), but it fails to account for important lifestyle factors such as diet, smoking, physical activity, and stress levels; which have a pivotal role in dylipidemia.

2. Likewise, mental health issues (such as anxiety, depression, insomnia, fatigue) also play a key role and they are not included as well (though mentioned in limitation)

3. No data about the medication use by healthcare workers, which might affect lipid levels.

4. A causal inference cannot be drawn through observational studies. Dyslipidemia risk cannot be decided based on the data in the manuscript.

5. Line 320 “Healthcare workers responsible for diagnosing and treating COVID-19 patients face more lockdowns than other populations, so their blood lipids will also increase”. is contradictory to the hypothesis and the results.

6. The study propose that high workload may have led to lower energy intake and better lipid profiles, but this assumption is speculative. Other possible explanations, such as selection bias (i.e., healthier workers were more likely to continue working in frontline roles), have not been explored. Additionally, high workload workers need energy and they might be taking high calorie diet. How this paradox is addressed.

7. The hazard ratios for some key variables like triglycerides and LDL cholesterol are statistically significant but lack clinical interpretation, whether this implies a significant health risk.

8. The use of stepwise regression for variable selection is controversial. This method can lead to overfitting and unstable models, especially in datasets with multicollinearity. Methods like LASSO or Ridge regression are more suitable in this case.

9. The study performs multiple statistical tests without adjusting for multiple comparisons, increasing the risk of Type I errors (false positives). Techniques like Bonferroni correction or False Discovery Rate Control are more appropriate.

10. What is meant by authorized strength.

11. It is not clear whether 'resignation' is exclusion or endpoint.

12. There are lot of lexical incoherence and the reader is confused. For example "Of the 67 dyslipidemia medical workers, 28 (3.8%) , 204 frontline medical personnel occurred in 2019-2020, 23 (3.2%) occurred in 2020-2021, and 16 (2.2%) occurred in 2021-2022” . This is observation not occurrence. Likewise "edge elevation" is not a scientific word. Manuscript needs to be checked for quality of English.

7. PLOS authors have the option to publish the peer review history of their article (what does this mean? ). If published, this will include your full peer review and any attached files.

**Do you want your identity to be public for this peer review?** For information about this choice, including consent withdrawal, please see our Privacy Policy .

Reviewer #2: No

Reviewer #3: **Yes: ** Dr Misbahuddin Rafeeq

---

## [Author Response · Author response to Decision Letter 2]

2 Mar 2025

Reviewer #3: The authors addressed an important issue regarding the factors associated with dyslipidemia in healthcare workers in a COVID-19- designated hospital in China. Though the topic is important but the study has major flaws. Some are mentioned in limitations by the authors as well.

Some of the issues that need to be addressed are

1. The study adjusts for several potential confounders in the multivariate Cox proportional hazard regression model (e.g., age, gender, frontline working time, biochemical parameters), but it fails to account for important lifestyle factors such as diet, smoking, physical activity, and stress levels; which have a pivotal role in dyslipidemia.

Answer: We would like to thank you for your critical and constructive comments. We 100% agree with your suggestion that lifestyle factors play a pivotal role in dyslipidemia. However, these factors were not evaluated during the pandemic because many of our medical staff focused on preventing and treating COVID-19. For example, my daughter was born during the COVID-19 era in January 2022, but I couldn't see her for six months because of the pandemic even if she entered the ICU. Although we know that lifestyle factors are important, we currently can’t evaluate their lifestyle factors. However, we added the lifestyle factors in the limitation with the following paragraph, wishing you and other readers can understand. “Thirdly, some other important variables, such as the psychological status, lifestyle factors, and medication used by the medical staff, should be included in future research.”

2. Likewise, mental health issues (such as anxiety, depression, insomnia, fatigue) also play a key role and they are not included as well (though mentioned in limitation)

Answer: We appreciate your constructive comments. The mental health also plays a key role in dyslipidemia. This was a retrospective cohort study using healthcare workers’ yearly physical examination results. Unfortunately, the psychological issues were not evaluated thus the mental health data was not available. We plan to address this limitation in our future research by studying the mental health of medical staff after COVID-19. We believe that your suggestion will further strengthen our research direction and contribute to the field.

3. No data about the medication use by healthcare workers, which might affect lipid levels.

Answer: Thank you very much for your insightful comments and constructive feedback on our manuscript. We truly appreciate the time and effort you have invested in reviewing our work.

Regarding the issue you raised about the medication used by healthcare workers, we fully acknowledge that this is a significant limitation of our study. Unfortunately, this defect is not something we can address within the scope of the current study. The primary reason is the data unavailability.

We understand that this limitation may affect the comprehensiveness of our findings. However, we would like to emphasize that our study still provides valuable insights into how frontline working time affects healthcare workers' dyslipidemia.

Moving forward, we plan to address this limitation in our future research by studying the long-term health of medical staff after COVID-19. We believe that this question and what you mentioned in questions 1 and 2 will further strengthen our research direction and contribute to the field.

Once again, thank you for your valuable comments. We have revised our manuscript to clearly state this limitation and have added a detailed discussion on its potential impact and our future plans.

4. A causal inference cannot be drawn through observational studies. Dyslipidemia risk cannot be decided based on the data in the manuscript.

Answer: Thank you for your attention and valuable comments on our research. The causal inference problem you mentioned is indeed an important scientific consideration, especially in observational research. We understand your concerns and elaborate on relevant issues here.

On causal inference

Observational studies, including retrospective cohort studies, do have some limitations, especially in causal inference. However, this does not mean that observational studies cannot provide valuable causal clues

In our study, we used some statistical methods to control for potential confounders, such as multivariable regression analysis to adjust for known confounders (such as age, sex, etc.), so as to minimize the impact of confounding bias on the results

Although observational studies cannot completely replace the role of randomized controlled trials (RCTs) in causal inference, RCTs are often limited by ethics, cost and feasibility in practical research. Therefore, observational research is still one of the important means to explore causality.

Determination of the risk of dyslipidemia

We acknowledge that it is indeed difficult to fully identify risk factors for dyslipidemia based on data from observational studies alone. However, our results provide important clues for the risk factors of dyslipidemia in front-line medical staff during COVID-19.

In addition, we also highlighted the limitations of the study in the discussion section, including the possible existence of unrecognized or uncontrolled confounding factors

Therefore, we suggest that future studies can further verify the causality of these associations through more rigorous experimental design (such as RCT) or more complex statistical methods (such as instrumental variable analysis)

5. Line 320 “Healthcare workers responsible for diagnosing and treating COVID-19 patients face more lockdowns than other populations, so their blood lipids will also increase”. is contradictory to the hypothesis and the results.

Answer: Thank you for your attention and valuable comments on our research. Our result demonstrated that the HRs of TG and LDL were 3.81 (2.11-6.86) (P<0.001), and 2.83 (1.64-4.87) (P<0.001), respectively. Medical staff with high TG and LDL at baseline were more likely to have dyslipidemia during the pandemic. But the sentence “Healthcare workers responsible for diagnosing and treating COVID-19 patients face more lockdowns than other populations, so their blood lipids will also increase” indeed caused some contradictions. Therefore, we rewrite this paragraph with the following paragraph “Medical staff with high TG, LDL, and low HDL at baseline were more likely to have dyslipidemia during the pandemic.”

6. The study propose that high workload may have led to lower energy intake and better lipid profiles, but this assumption is speculative. Other possible explanations, such as selection bias (i.e., healthier workers were more likely to continue working in frontline roles), have not been explored. Additionally, high workload workers need energy and they might be taking high calorie diet. How this paradox is addressed.

Answer: Thank you for your attention and valuable comments on our research. After carefully considering your suggestion, we decided to rewrite the content of this paragraph with the following paragraph:

“There were three possible explanations. Firstly, the COVID-19 pandemic put extraordinary pressure on healthcare workers both physically and psychologically (22, 23). The heavy workload will keep healthcare workers physically active, thus leading to an increase in energy consumption. However, missing families, anxiety, depression, and other mental problems can undermine the appetite of healthcare workers and thus lead to a decrease in energy intake (9, 10). Our results suggest that the outcome of the paradox was that mental health may have a more significant impact on energy intake. A study (24) from Hubei Province conducted from March 19 to April 1, 2020, found that 22.9% of healthcare workers reported weight loss, and more than 50% of healthcare workers asked for a balanced diet, including more coarse grains, vegetables, nuts, fruits, and soybeans. Secondly, most healthcare workers who entered the frontline were physically and psychologically healthy, so there may be selection bias. Research (25) also found that medical staff who maintain their self-regulatory eating behavior are more likely to be stress-free. Thirdly, those medical personnel who have not entered the front line, have more time with families, and a more relaxed environment to consume delicious food, which to some extent can also lead to a higher lipid profile. Therefore, this study found that medical staff with a high (≥ 30 days) frontline working time have a lower risk of developing dyslipidemia.”

7. The hazard ratios for some key variables like triglycerides and LDL cholesterol are statistically significant but lack clinical interpretation, whether this implies a significant health risk.

Answer: Thank you for your attention and valuable comments on our research. we add the following paragraph into the manuscript to state their clinical significance.

“However, elevated levels of LDL are an independent risk factor for atherosclerotic cardiovascular disease (ASCVD). The higher the LDL levels, the greater the risk of cardiovascular events (33), such as coronary heart disease, myocardial infarction, and stroke. While TG is not an independent risk factor for coronary heart disease, the presence of elevated TG levels in combination with other abnormalities, such as high LDL and low HDL, significantly increases the risk of cardiovascular disease.”

8. The use of stepwise regression for variable selection is controversial. This method can lead to overfitting and unstable models, especially in datasets with multicollinearity. Methods like LASSO or Ridge regression are more suitable in this case.

Answer: Thank you for highlighting the controversies surrounding variable selection in stepwise regression, as well as the potential issues of overfitting and model instability, especially in datasets with multicollinearity. Your perspective is entirely correct, and we fully understand and agree with these points.

To address these concerns, we employed LASSO regression to screen for the univariate variables, using the regularization parameter λ corresponding to lambda.min (0.006053774). Ultimately, eight variables were selected, but WBC was deleted because of no significant change of the AIC.

Additionally, to compare the models selected by stepwise regression with those derived from LASSO, we conducted the following comparison. Once again, thank you for your valuable comments.

Model AIC P

Model before revise 791.09

Model revised with your suggestion 784.76 0.006

9. The study performs multiple statistical tests without adjusting for multiple comparisons, increasing the risk of Type I errors (false positives). Techniques like Bonferroni correction or False Discovery Rate Control are more appropriate.

Answer: We are extremely grateful for your suggestion. We have indeed conducted numerous subgroup analyses to compare the characteristics of frontline working time across different groups. However, in each subgroup analysis, we have adjusted for all the remaining variables. We understand that your concern primarily stems from the lack of clarity in our methodological description. Therefore, we have added the following paragraph to the methodology section:

"In each subgroup analysis, we adjusted for all remaining variables to account for the impact of frontline working time on the occurrence of dyslipidemia."

10. What is meant by authorized strength.

Answer: Thank you for your valuable comments on our research. The “authorized strength” in Chinese hospitals means that the positions are often supported by government funding and provide stable employment opportunities. After careful consideration, we decided to use “establishment positions” to replace “authorized strength”, which could be more appropriate. The number of “establishment positions” is limited and determined by the government based on the scale and functional requirements of the hospital. These establishment positions offer better employee income, comprehensive benefits, opportunities for career development, and enhanced social status.

11. It is not clear whether 'resignation' is exclusion or endpoint.

Answer: Thank you for your valuable comments on our research. Resignation was defined as loss to follow-up. There were two types we deal with the resignation: (1) medical staff had dyslipidemia before the resignation, and this data was complete data; (2) medical staff didn’t have dyslipidemia before the resignation, and this data was censored data. All the data used in our study was corrected by the censored data.

12. There are lot of lexical incoherence and the reader is confused. For example "Of the 67 dyslipidemia medical workers, 28 (3.8%) , 204 frontline medical personnel occurred in 2019-2020, 23 (3.2%) occurred in 2020-2021, and 16 (2.2%) occurred in 2021-2022” . This is observation not occurrence. Likewise "edge elevation" is not a scientific word. Manuscript needs to be checked for quality of English.

Answer: Thank you for your valuable comments on our research.

There were two reasons to explain why we used the observation, not the occurrence. Firstly, numerous studies have investigated the work stress experienced by medical staff during the COVID-19 pandemic, often employing questionnaires to assess their intentions to resign. However, our study stands out by utilizing the most authentic data. We aim to present this data most intuitively to our readers, providing them with an understanding of the actual resignation data of medical personnel in designated COVID-19 hospitals. Secondly, we specifically highlight the periods of 2019–2020, 2020–2021, and 2021–2022 in our presentation. We added some explanations in the methodology section for the three periods with the following paragraph. “There were three periods: 2019-2020 means the time of medical staff from the time of the 2019 physical examination to the 2020 physical examination, and so on 2020-2021, and 2021-2022. These periods correspond to two major quarantine periods in our hospital. We found that the number of medical staff resignations was the highest during these two major quarantine periods, which we also want to demonstrate to our readers. The medical staff in designated COVID-19 hospitals face significant stress. For example, my daughter was born during the COVID-19 era in January 2022, but I couldn't see her for six months because of the pandemic even if she entered the ICU. I also contemplated resigning and began submitting applications to other hospitals. However, due to certain special circumstances, I ultimately did not leave my current position.

After careful consideration, we decided to use “slightly elevated” to replace “edge elevation”, which could be more appropriate.

We have carefully reviewed our manuscript and made the necessary revisions to address the language issues you pointed out. We have also engaged a professional language editor to ensure that the English is clear, concise, and free of errors.

---

## [Decision Letter · Decision Letter 2]

Factors associated with dyslipidemia among healthcare workers in a COVID-19-designated hospital in Hangzhou, Zhejiang, China: a retrospective cohort study from 2019 to 2022

PONE-D-24-13033R2

Dear Dr. Xu,

We’re pleased to inform you that your manuscript has been judged scientifically suitable for publication and will be formally accepted for publication once it meets all outstanding technical requirements.

Kind regards,

Colin Johnson, Ph.D.

Academic Editor

PLOS ONE

Additional Editor Comments (optional):

Reviewers' comments:

Reviewer's Responses to Questions

**Comments to the Author**

1. If the authors have adequately addressed your comments raised in a previous round of review and you feel that this manuscript is now acceptable for publication, you may indicate that here to bypass the “Comments to the Author” section, enter your conflict of interest statement in the “Confidential to Editor” section, and submit your "Accept" recommendation.

Reviewer #3: All comments have been addressed

2. Is the manuscript technically sound, and do the data support the conclusions?

Reviewer #3: Yes

3. Has the statistical analysis been performed appropriately and rigorously? 

Reviewer #3: Yes

4. Have the authors made all data underlying the findings in their manuscript fully available?

Reviewer #3: Yes

5. Is the manuscript presented in an intelligible fashion and written in standard English?

Reviewer #3: Yes

6. Review Comments to the Author

Reviewer #3: Kindly include the revised comments addressing the key limitations of the manuscript in a separate "limitations" section

7. PLOS authors have the option to publish the peer review history of their article (what does this mean? ). If published, this will include your full peer review and any attached files.

**Do you want your identity to be public for this peer review?** For information about this choice, including consent withdrawal, please see our Privacy Policy .

Reviewer #3: **Yes: ** misbah

---

## [Editor Report · Acceptance letter]

PONE-D-24-13033R2

PLOS ONE

Dear Dr. Xu,

I'm pleased to inform you that your manuscript has been deemed suitable for publication in PLOS ONE. Congratulations! Your manuscript is now being handed over to our production team.

Kind regards,

on behalf of

Dr. PLOS Manuscript Reassignment

Staff Editor

PLOS ONE